# Strain-Level Dynamics Reveal Regulatory Roles in Atopic Eczema by Gut Bacterial Phages

Yanan Chu,[a] Qingren Meng,[b] Jun Yu,[c] Juan Zhang,[d] Jing Chen,[a] Yu Kang[a]

[a]Beijing Institute of Genomics, Chinese Academy of Sciences/China National Center for Bioinformation, Beijing, China
[b]School of Medicine, Southern University of Science and Technology, Shenzhen, China
[c]University of Chinese Academy of Sciences, Beijing, China
[d]Department of Pediatric, Peking University Third Hospital, Beijing, China

Yanan Chu and Qingren Meng contributed equally. Author order was determined alphabetically by last name.

**ABSTRACT** The vast population of bacterial phages or viruses (virome) plays pivotal roles in the ecology of human microbial flora and health conditions. Obstacles, including poor viral sequence inference, strain-sensitive virus-host relationship, and the high diversity among individuals, hinder the in-depth understanding of the human virome. We conducted longitudinal studies of the virome based on constructing a high-quality personal reference metagenome (PRM). By applying long-read sequencing for representative samples, we could build a PRM of high continuity that allows accurate annotation and abundance estimation of viruses and bacterial species in all samples of the same individual by aligning short sequencing reads to the PRM. We applied this approach to a series of fecal samples collected for 6 months from a 2-year-old boy who had experienced a 2-month flare-up of atopic eczema (dermatitis) in this period. We identified 31 viral strains in the patient's gut microbiota and deciphered their strain-level relationship to their bacterial hosts. Among them, a lytic crAssphage developed into a dozen substrains and coordinated downregulation in the catabolism of aromatic amino acids (AAAs) in their host bacteria which govern the production of immune-active AAA derivates. The metabolic alterations confirmed based on metabolomic assays cooccurred with symptom remission. Our PRM-based analysis provides an easy approach for deciphering the dynamics of the strain-level human gut virome in the context of entire microbiota. Close temporal correlations among virome alteration, microbial metabolism, and disease remission suggest a potential mechanism for how bacterial phages in microbiota are intimately related to human health.

**IMPORTANCE** The vast populations of viruses or bacteriophages in human gut flora remain mysterious. However, poor annotation and abundance estimation remain obstacles to strain-level analysis and clarification of their roles in microbiome ecology and metabolism associated with human health and diseases. We demonstrate that a personal reference metagenome (PRM)-based approach provides strain-level resolution for analyzing the gut microbiota-associated virome. When applying such an approach to longitudinal samples collected from a 2-year-old boy who has experienced a 2-month flare-up of atopic eczema, we observed thriving substrains of a lytic crAssphage, showing temporal correlation with downregulated catabolism of aromatic amino acids, lower production of immune-active metabolites, and remission of the disease. The PRM-based approach is practical and powerful for strain-centric analysis of the human gut virome, and the underlying mechanism of how strain-level virome dynamics affect disease deserves further investigation.

**KEYWORDS** virome, strain level, dynamics, crAssphage

Address correspondence to Yu Kang, kangy@big.ac.cn.

The authors declare no conflict of interest.

[This article was published on 23 March 2023 with missing information in Acknowledgments. The Acknowledgments were corrected in the current version, posted on 27 March 2023.]

The viruses or bacteriophages dwelling in the human intestines are essential components of the entire microbial flora, whose numbers as particles often override those of their bacterial hosts (1, 2). Growing evidence has associated the phageome with pathogeneses of many human diseases, such as inflammatory bowel diseases (3), obesity and type 2 diabetes mellitus (4), malnutrition metabolic syndrome (5, 6), and autoimmune disease (7, 8), via regulating the gut microbial metabolism and host immunity (9, 10). However, these viruses are still regarded as "dark matters of the guts," mainly due to the limited knowledge about their taxonomy, bacterial hosts, and roles in ecological relatedness between microbial flora and human health. One of the many obstacles in understanding these viruses is the poor homology and extensive recombination of their genomes (11), which substantially confound the identification of their genomes from metagenomic data and distinct assignment to a specific taxon (12, 13). The huge strain diversity in virome profiles, i.e., very few viruses are shared among human individuals, makes it even more improbable to compare a given virus across human individuals (1, 13). Furthermore, the receptor-induced viral infection of host bacteria makes the virus-host interactions strain sensitive (14–18). However, until now, virome analysis with strain-level resolution has not yet been reported, making the role of viruses in human health and disease even more challenging to be elucidated.

Here, we propose a new approach leveraging the construction of a high-quality personal reference metagenome (PRM) that provides strain-level identities for analyzing the virome. As the virome often persists within a given human host for a relatively long period (13), longitudinal samples are essential to establish dynamic relationships between the virome, the bacteriome, and host health conditions. A high-quality nonredundant PRM, which contains sequence assemblies of all viruses and bacteria in all the longitudinal samples, can ensure the accurate abundance estimation of each viral strain in the entire microbiota instead of in the virome by mapping short-read sequences to it. Our approach for building the PRM is based on the long-read sequencing of representative samples, which provides contiguity for accurate taxonomical annotations of each assembled contig and avoids substantial redundancy. As such, the PRM-based analysis is a resolution for the strain-level distinction of viral genomes and the description of their variations and dynamics in the context of individual human microbial flora. When we applied this approach to a series of fecal samples from a 2-year-old boy who had experienced a flare-up of acute dermatitis (eczema) and recovered during the observation period, the dynamics of each virus and its host exhibited strain-level sensitivity and ecological relationship, where the upsurge of various crAssphage substrains coincided with disease remission of the infant patient.

## RESULTS

**The PRM improves virome annotation.** To construct the personal reference metagenome (PRM) for an individual patient in a longitudinal study, we initiated the assembly with one or two representative samples applied to long-read sequencing, from either the nanopore platform (19) or the PacBio platform (20), and subsequently added low-cost short-read sequences of all other samples, generated from the Illumina platform (Fig. 1A). The long-read assembly provides satisfactory contiguity for viral genomes, and the average length of these PRM contigs often reaches several hundred kilobase pairs according to our previous experience, long enough for accurate viral genome identification and taxonomic assignment of bacterial sequences, as well as gene annotation.

To test the performance of the PRM, we have applied serial fecal samples collected from a 2-year-old boy over 170 days from January to July 2019. During this period, the boy experienced a flare-up of severe eczema, which lasted for about 2 months and was relieved naturally upon diarrhea that lasted a week without specific treatments. A total of 24 samples were used for the study, where 11 samples were from the active phase (AP) and 2 and 11 remission-phase (RP) samples were from days before and after the active phase, respectively (see Table S1 in the supplemental material). The severity of eczema was indexed according to the Eczema Area and Severity Index (EASI) (21) upon

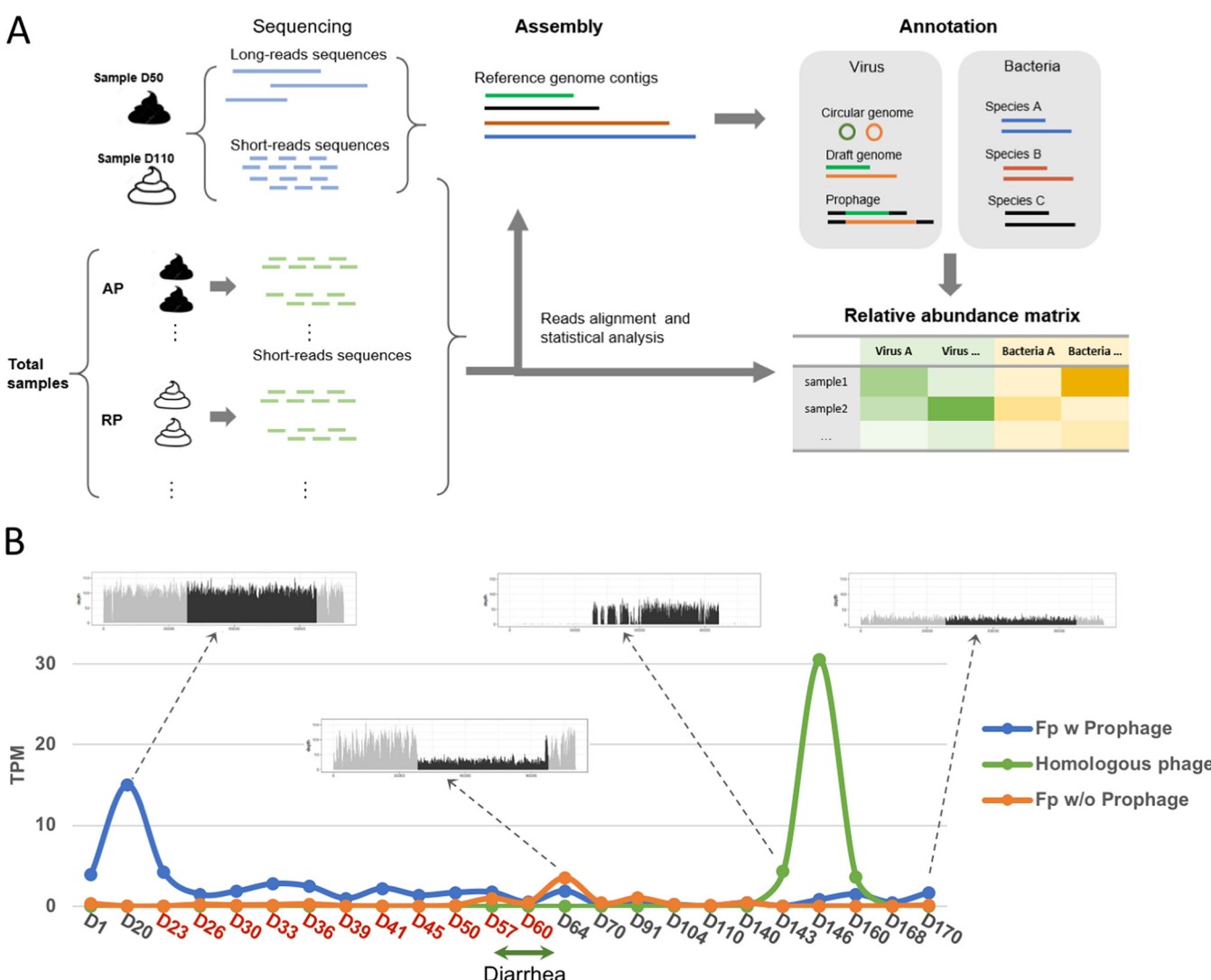

**FIG 1** The personal reference metagenome (PRM) and virome analysis at the strain level. (A) A schematic illustration of PRM construction and its application in virome analysis. AP, active phase; RP, remission phase. (B) The variation dynamics of F.prausnitzii-prophage-63kb and its host bacteria during the observation period. The vertical axis displays TPM of the F.prausnitzii-prophage-63kb, its flanking bacterial sequences, and another homologous phage strain to indicate their abundance in each sample. The horizontal axis indicates sampling days, where timing of AP and RP is highlighted in red and black, respectively. The boxes show mapped contigs and reads from representative samples. Fp, *F. prausnitzii*; w, with; w/o, without.

sample collection (Table S2). Two samples (D50 and D110 in the AP and RP group, respectively) were used for nanopore sequencing, and all samples were applied for paired-end (150-bp) shotgun sequencing on the Illumina platform (Fig. 1A; Table S1).

To obtain the PRM, we assembled the nanopore sequences and shotgun sequences from the two representative samples separately and removed redundant contigs. The final PRM of the boy contained 5,431 contigs and was 141 Mb in total length ($N_{50}$ = 71.8 kb), much longer than those of contigs assembled solely based on short reads (Table S3) and sufficient for confident viral sequence identification. By utilizing the VIBRANT tool for viral sequence annotation (22), we identified 234 virus fragments, among which 31 viral genomes (strains) were confirmed by both VirSorter and vConTACT2 (Fig. S1), including nine circular virus complete genomes (with their genome structure shown in Fig. S2), nine candidate genomes with high or medium confidence according to the rules of VIBRANT for evaluating virus genome quality (22), and 13 prophages integrated into bacterial genomes (Table S4). The quality of the 31 viral genomes evaluated by checkV exhibited consistency with VIBRANT (Table S4). We have further categorized the viruses as lytic or lysogenic, predicted their bacterial host based on CRISPR spacer sequences for phages,

and reannotated the host of prophages in our samples by annotating their flanking bacterial sequences. Each phage strain in our samples is named in the same way as "host-phage/prophage-length," with its bacterial host labeled as the prefix and its genome size as the suffix, respectively. All the viruses in our samples are phages, and there has been no virus of eukaryotic origin identified. The genome length of these phages ranges from 6 to 158 kb with an average of 40 kb. Among the 31 phages, 12 are lytic, exhibiting a slightly lower GC content of 41.4% $\pm$ 10.3%, compared to 47.3% $\pm$ 7.6% of lysogenic phages. Other contigs of bacterial origin were assigned to 460 bacterial/archaeal species (Fig. S1).

When mapping the short reads of each sample to the PRM, we achieved an average mapping rate of as high as 99.10% $\pm$ 0.52%. Such an unprecedented mapping rate indicates the PRM can represent the microbial DNA in the samples with negligible loss. The relative abundance of all viruses in our samples ranged from 1.36 to 45.90% with an average of 6.53%. When calibrated by the average genome size (40 kb [4 Mb]), the total relative abundance of assembled phages indicates a rough ratio between the viral and bacterial particles of 9.29:1, roughly consistent with the previous reports based on images from electron microscopy (1, 2, 23).

Metagenome-assembled genome (MAG) reconstructions from shotgun metagenomic sequence data are another approach to obtaining the PRM that needs no additional long-read sequencing (24–26). Here, we also tried this strategy by binning the short-read assembled contigs of each sample using MetaWRAP. Then, we pooled all the bins from the 24 samples and removed redundancy using dRep to obtain the MAG-based PRM. Compared with our long-read-based PRM, the MAG-based PRM performed worse at virus identification, especially for the circular or high-quality genomes, due to the relatively short length of its contigs (Table S5). The final MAG-based PRM also lost many sequences even after pooling bins from all the samples due to the trimmed contigs in the binning process, which leads to a significantly lower mapping rate of 85.55% $\pm$ 0.04%, indicating its poorer representativeness.

**The PRM allows strain-level analysis for assessing virome dynamics.** Our results in deciphering viral strain dynamics show clear in-and-out events during the observation (Fig. S3), including complete depletion of the Streptomyces-proPhage-57kb by its host (Fig. S4A) and the surge of the lytic crAssphage-99kb (Fig. S4B). Interesting strain-level phage-host dynamics also imply a complex ecological relationship among viruses and their bacterial hosts. The relative abundance of F.prausnitzii-prophage-63kb compared with that of its host coincides initially but separates after diarrhea (Fig. 1B). Comparing the mapped reads to the contig containing the prophage and flanking host bacterial sequences, we are able to identify intriguing strain substitutions in both the phage and its host. In the first phase, a strain of *Faecalibacterium prausnitzii* carrying the F.prausnitzii-prophage-63kb appears initially, but upon diarrhea, another *F. prausnitzii* strain without the prophage overrides the initial strain in number for just a few days. The abundance of the phage and its host strain decreases to a deficient level until 12 weeks after diarrhea, when another phage homolog to the F.prausnitzii-prophage-63kb with an overall genome identity of 95.6% (Fig. S4C) is introduced and reaches a very high abundance for a dozen days. Finally, the abundance of the initial *F. prausnitzii* strain carrying the F.prausnitzii-prophage-63kb recovers to its original level (Fig. 1B). The dynamics of their abundance depict the complex process of competition between viruses and their host bacterial strains in an actual human metagenome sample.

Substrain resolution can also be obtained in some cases. Comparing the reads mapped to the crAssphage-99kb genome, we have identified neither any sequence indels nor recombination events but found tens of single nucleotide polymorphisms (SNPs) that concentrate within a narrow window of 75 bp in length in the coding sequence of a putative tail fiber protein, and the variations are all nonsynonymous types, i.e., they lead to alteration of amino acid sequences (Fig. 2A). According to these aligned SNPs in the tail protein, there have been 11 substrains identified among the phage variants. We observed the abundance dynamics of crAssphage-99kb and its top three most abundant substrains, as well as the count of substrains in each sample (Fig. 2B). Not only is there coexistence of

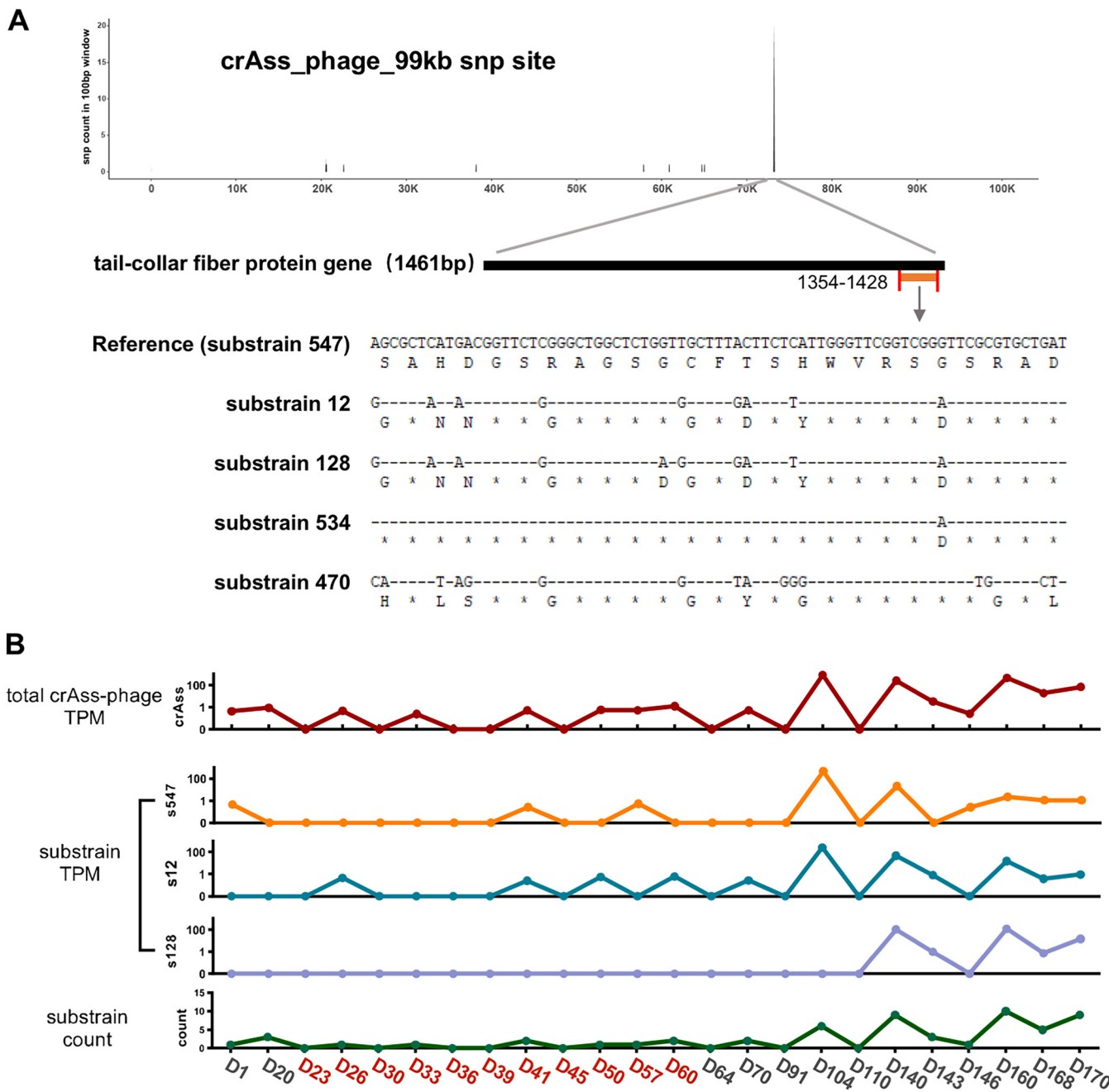

**FIG 2** The upsurge of crAssphage-99kb substrains. (A) The upper panel is the number of SNPs in each 100-bp sliding window of the crAssphage-99kb genome. The tail-collar fiber protein gene is enlarged to show highly enriched nonsynonymous variations at the end of the sequence. The lower panel depicts the main substrains and their protein sequences at the end of the tail-collar fiber protein. (B) The TPM of crAssphage-99kb and its three major substrains as well as the count of the substrains. The horizontal axis indicates the days when samples were collected, where the samples are divided into AP and RP, highlighted in red and black, respectively. AP, active phase; RP, remission phase.

crAssphage-99kb substrains found in the same sample, but also the substrains exhibit an evident trend where their numbers increase after diarrhea together with their abundance in total. These findings suggest that diarrhea triggers fast sequence diversification of the tail fiber protein gene, potentially broadening its host range and promoting its propagation in the flora (27).

**Upsurge of crAssphage substrains correlates with disease remission.** From the sequence variability of the gut virome, it appears that diarrhea triggers radical adjustment in fitness among phages and their hosts, and the event cooccurs with the spontaneous remission of eczema. Therefore, we compared the variation patterns of the virome

between AP and RP samples for their association with eczema. First, the abundance of phages in total sharply increases after diarrhea and lasted for a few months (Fig. 3A), and the incremental surges are mainly attributable to the enrichment of several lytic phages, including crAssphage-99kb (Fig. 3A and B and Fig. S5A). In contrast, the relative abundance of lysogenic phages decreases after diarrhea (Mann-Whitney U test, $P = 0.013$ [Fig. S5B]). The overall abundance of all lytic phages and crAssphage-99kb indicates a significant increase in the RP samples over the AP samples (Mann-Whitney U test, $P_{lytic} = 0.0012$, $P_{crAss} = 0.00098$ [Fig. 3B]). Six out of 31 phages show significant alteration in their abundance across the AP and RP samples, with four lysogenic phages reduced and two lytic phages increased, including crAssphage-99kb, which increases in the RP (Mann-Whitney U test, false-discovery rate [FDR] of $<0.05$ [Fig. S6]). Correlations between the phage abundance and EASI score (Spearman's rank correlation test, $\rho > 0.5$ or $<-0.5$ and $P < 0.01$) indicate that the lytic crAssphage_99kb and circular_phage_158kb show the strongest negative correlations with the EASI score, while a variety of prophages exhibit strong positive correlations (Table S6 and Fig. S7). These results suggest opposing changes between some lytic and lysogenic phages and their potential correlations with eczema pathogenesis.

We next focus on the metabolic coordination of crAssphage-99kb and its host bacterium because this high-abundance phage has experienced the most profound changes upon eczema remission, including the highly diversified tail protein gene, the upsurge of substrains, and sharp enrichment in the host. Although we have failed to identify the actual host at the species level of crAssphage-99kb in our samples based on the CRISPR spacer sequences, the candidate hosts of crAssphages have been reported previously (Table S7) (8, 28, 29) and are identified in our samples. As expected, most candidate bacterial hosts reduce their abundance in RP samples, with the relative abundance of *Clostridium botulinum*, *Bacillus cereus*, and *Fusobacterium nucleatum* declining significantly (Mann-Whitney U test, FDR $< 0.05$ [Table S7 and Fig. S8]). Many of the crAssphage hosts, including *Clostridium botulinum*, *Bacillus cereus*, *Fusobacterium nucleatum*, *Blautia* sp., *Anaerobutyricum hallii*, *Blautia producta*, *Bacteroides fragilis*, *Ruminococcus* sp., *Parabacteroides distasonis*, and *Bacteroides uniformis*, can metabolize aromatic amino acids (AAAs; including tyrosine, tryptophan, and phenylalanine) (30–33). We also observed fluctuations in the relative abundance of the three AAA-metabolizing bacteria following the pace of the crAssphage-99kb fluctuation in our samples (Fig. 3C), and the total abundance of the AAA-metabolizing bacterial hosts significantly appeared to decline in the RP (Mann-Whitney U test, $P = 0.035$ [Fig. 3D]).

According to the microbial AAA-catabolism pathways recently clarified (34), AAAs share a set of enzymes in which genes *fldH*, *fldB*, and *fldC* are essential for the reductive metabolism of all three AAAs. A variety of AAA derivatives are immune active and reported to participate in immune regulation (34–36). Benefiting from the accurate annotation of the PRM for genes, we were able to identify all *fldH*, *fldB*, and *fldC* genes in all bacterial species of our samples by constructing hidden Markov models (HMM) for these genes (37). The relative abundance of these genes exhibits synchronism with that of AAA-metabolizing bacteria and also lags behind the fluctuation of crAssphage-99kb (Fig. 3C). The total transcripts per million (TPM) of all copies of *fldH*, *fldB*, and *fldC* are all significantly reduced after remission (Mann-Whitney U test, $P_{fldH} = 0.000054$, $P_{fldB} = 0.0031$, $P_{fldC} = 0.0025$ [Fig. 3D]). These findings suggest a complex predator-prey relationship, as well as the potential regulatory mechanism of crAssphage-99kb for the abundance of its hosts and the metabolic contribution of these bacterial species to gut microbial flora.

To confirm the metabolic change predicted from virome and metagenome analyses, we have also identified nine metabolites that significantly changed between AP and RP samples, using the nontargeted metabolome assay, and among them, five are catabolic metabolites of AAAs, increased in AP (Mann-Whitney U test, FDR $< 0.05$ [Fig. 3E and Fig. S9]). These differential AAA metabolites include those of immune-active indolepropionic acid and indole-3-methyl acetate (34–36), and some have been reported to be associated with various inflammation-associated diseases, including phenyllactic acid and 4-OH-phenylpropionic acid (30, 38–40). This result supports a

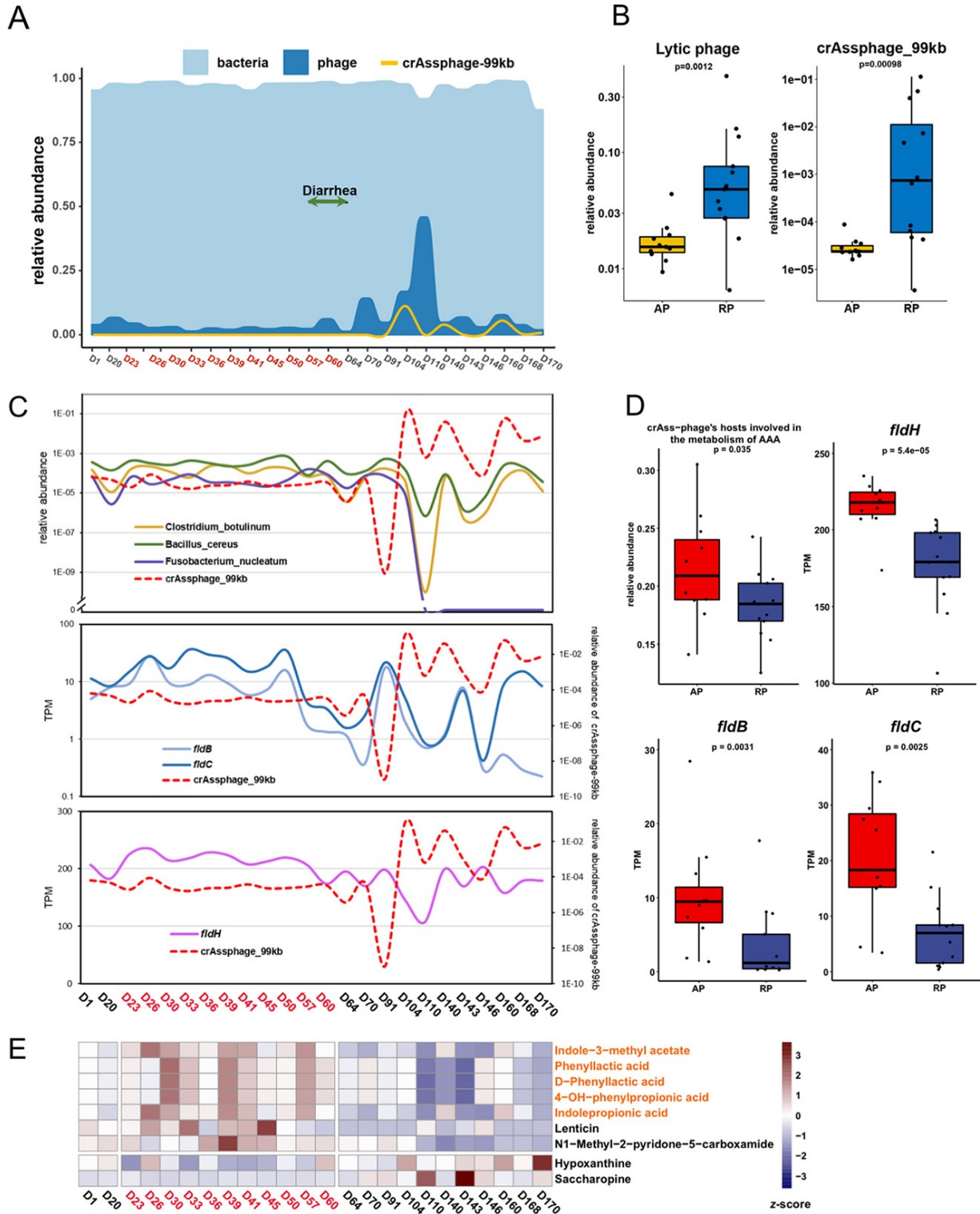

**FIG 3** The variation of the virome between the AP and RP groups. (A) The abundance of total bacterial species, phages, and crAssphage-99kb. (B) The relative abundance of lytic phages and crAssphage-99kb between AP and RP samples. (C) Relative abundance of AAA-metabolizing candidate host of crAssphage-99kb (upper panel) and TPM of AAA-metabolizing essential genes (lower panel) in contrast to the fluctuation in the abundance of crAssphage-99kb. (D) The comparison of AAA-metabolizing candidate hosts of crAssphage-99kb and AAA-metabolizing essential genes between samples from AP and RP. (E) The heatmap of relative concentrations of metabolites that significantly altered between samples from AP and RP (Mann-Whitney U test, FDR < 0.05). The horizontal axis depicts days when samples were collected from AP and RP, in red and black, respectively. AP, active phase; RP, remission phase. The significance of differences in abundance or TPM between groups was tested with the Mann-Whitney U test.

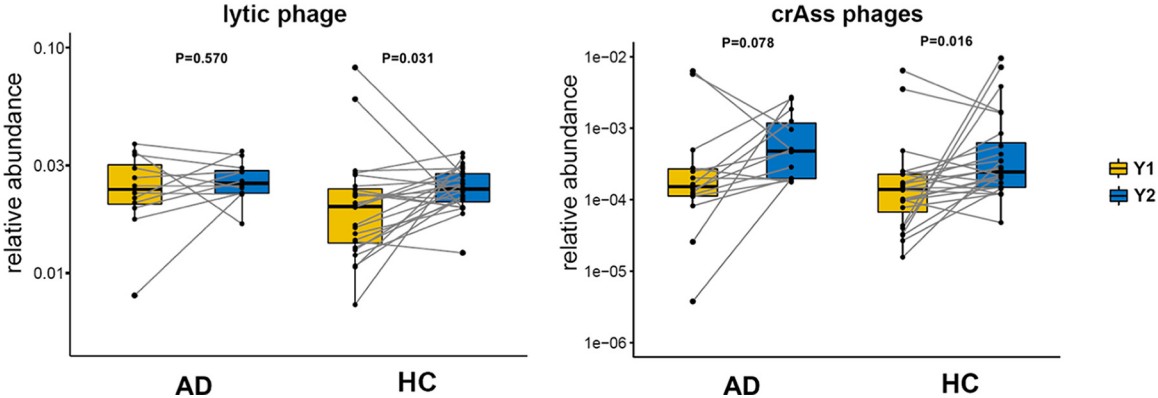

**FIG 4** Relative abundance of lytic phages and crAssphages in fecal samples collected at the age of 1 (Y1) and 2 (Y2) years based on raw data from a published study of infant eczema. The cases contain infants who experienced eczema before 2 years of age with atopic dermatitis (AD), and the controls are healthy subjects (HC). The significance of differences in abundance between groups was tested with the Mann-Whitney U test.

potential regulatory role of the crAssphage in the metabolism of the gut flora among atopic eczema patients. The relative concentrations of these immune-active AAA metabolites also show strong positive correlations with the EASI (Spearman's rank correlation test, Table S8). Therefore, the downregulated AAA catabolism potentially caused by upsurges of crAssphage-99kb substrains implies a new mechanism of how phages regulate gut metabolism, and its consequent influence on host bacteria deserves further investigation. However, the short-chain fatty acids, which are also reported as immune active (41, 42), showed no difference between AP and RP samples in our quantitative assay (Fig. S10).

**The hypoabundance of crAssphages and other lytic phages is further confirmed by mining public metagenomic data on eczema patients.** To further validate the role of crAssphage or other lytic phages in eczema, we have collected all three published metagenomic data sets of the infant eczema cohorts (43–45). Unfortunately, no samples had been collected during the acute phase of eczema for the atopic eczema subjects in all the reported studies. Two of the studies collected cross-sectional cohorts, and only one (45) had longitudinal metagenomic and metadata for gut samples from each subject. For all subjects in this longitudinal study, their fecal samples were collected at the age of 1 and 2 years, but the severity of eczema at the time of collection was not specified.

Here, we have used the data from two groups in the studies where the patients were treated with a placebo, and one of the groups experienced infant eczema before reaching the age of 2 years (atopic dermatitis [AD], $n = 14$) and the other had not (healthy controls [HC], $n = 24$). All other groups were treated with probiotics and not utilized for our analysis. Without the help of the PRM, we could only estimate the abundance of crAssphage and other lytic phages broadly by mapping reads to the GPD database (12), and they showed no significant differences between the AD and HC groups. However, when we compared the abundance increment from 1 to 2 years in age, the results showed that HC subjects appeared to have experienced a significant increase of lytic phages (Mann-Whitney U test, $P = 0.031$) and crAssphages (Mann-Whitney U test, $P = 0.016$) at the age of 2 years compared to samples collected at 1 year of age, while the AD group did not (Fig. 4), which was broadly consistent with our findings.

## DISCUSSION

Our study illustrates the reliable dynamics of each viral strain in the context of a human gut microbial flora. The dynamics of both viruses and their host bacteria indicate that different strains, even substrains with high homology, exhibit entirely different patterns in host-based and strain- or substrain-associated enrichment, as shown by examples of F.prausnitzii-prophage-63kb and crAssphage-99kb. The strain-specific sequence dynamics are consistent with the knowledge that a successful phage infection depends

on its ligand and receptors on host bacteria (15–18), making the predator-prey relationship strain sensitive. The strain-sensitive effects of phages on the ecology of microbial flora, together with the rapid mutation and recombination of phage genomes (11, 46), make the investigations of the virome very difficult. By constructing a high-quality reference metagenome by virtue of long-read sequencing platforms, we show that strain-resolution analysis of the virome in the context of the entire microbiome is achievable. The PRM, when applied to longitudinal samples of a single individual, enables accurate inference of the viral genome, host bacteria, and the genes they carry, as well as accurate abundance estimation in the entire microbiome instead of just in the virome by counting the short reads mapped to the PRM. Therefore, the PRM-based virome analysis should be an easily applied and cost-effective tool for high-resolution investigation of viruses related to human health and diseases.

With longitudinal samples from a 2-year-old boy who experienced a flare-up of eczema during our observational period, we illustrate the effectiveness of PRM, and the dynamics of relevant viruses in our samples show significant variabilities in abundance relative to their host, especially after diarrhea. Our results reveal fast-changing and complex phage-host interactions in human gut flora, especially after irregular events irritating the flora, such as diarrhea. Both the host range of phages and the population density of phages and their bacterial hosts may substantially impact the fitness of phage propagation. Meanwhile, the host range of phages may also change fast due to their fast genome evolution driven by mutation and recombination, as previously reported (11, 14, 27, 46). Therefore, longitudinal research is necessary for the insightful understanding of the human virome and the complex structure of the phage-host interactions.

One of our intriguing findings is the substrain upsurge of the lytic crAssphage-99kb, possibly due to the rapid diversification of its tail protein and expanded host range (27). The enrichment of crAssphage-99kb cooccurred with spontaneous eczema remission, which implies the potential benefit of this phage in soothing inflammation. Previous studies have reported that crAssphages are important components of the healthy human gut microbiota (47) and significantly decreased in the gut of patients with autoimmune diseases, such as inflammatory bowel disease, rheumatoid arthritis, and systemic lupus erythematosus (8, 28). These observations suggest a potential role of crAssphages in maintaining human health, especially in inhibiting gut inflammation. Our study also provides evidence that the downregulation of AAA-related catabolism is possibly due to the lytic effect of crAssphage-99kb on its AAA-metabolizing hosts. As metabolites from AAAs are well acknowledged to be immune active, these findings provide a potential mechanism for how phages regulate human immunity by regulating the metabolism of their hosts.

Although we have exciting findings from the single case, definite conclusions on the role of the virome, especially crAssphages, in maintaining the normal immune status of a human host are far from being established. In most of the cross-sectional studies on the virome, the huge strain diversity in virome profiles among human individuals makes it more difficult to compare a given virus across subjects, and the assemblies based on short-read sequences impair the annotation of the virome. Thus, we cannot fully validate our findings on the roles of the crAssphages and other phages in infant eczema due to the lack of public sequencing data from longitudinal observation of the gut virome, especially those of long-read sequencing. However, the complex strain-sensitive phage-host ecological relationship and the potential effects of virome alteration on human health that we observed in the single case highlight the importance of longitudinal studies and the merit of long-read-based sequencing in deciphering the pathogenesis of gut phages in various human diseases.

## MATERIALS AND METHODS

**Subject recruitment and sample collection.** We recruited a 2-year-old boy who suffered from severe food-induced atopic eczema after birth in Beijing, China. We collected 24 fecal samples from him during an atopic eczema flare-up and recovery from January to July 2019. During this period, the boy

was fed on a weekly fixed diet excluding probiotics and foods known to trigger allergies. In the phase of AD, the severity of eczema fluctuated and was scored according to the Eczema Area and Severity Index (EASI) (21). Fecal samples were frozen at −20°C on the day of collection and then transported to the Beijing Institute of Genomics, Chinese Academy of Sciences (CAS), and stored at −80°C within 2 days for subsequent metagenome DNA extraction and sequencing.

**DNA extraction, metagenome sequencing, and data quality control.** The total DNA of each fecal sample was extracted based on the E.Z.N.A. stool DNA kit (D4015; Omega) and the instructions from the manufacturer. Sequencing was performed on Illumina HiSeq X10 (150 bp by 2 bp; insert size, 350 bp) and GridION X5 platforms according to the information in Table S1 in the supplemental material. Raw reads from the Illumina HiSeq platform were next applied to quality control and filtered with Kneaddata (v0.7.2) to remove low-quality or contaminated human-genome reads.

**Annotation and construction of the personal reference metagenome.** For samples D50 and D110, clean reads from the Illumina HiSeq and GridION nanopore platforms were assembled using metaSPAdes (v3.15.2) for each sample. Total contigs with a length of >5,000 from two samples were clustered using CD-HIT-EST (v4.8.1; option: -c 0.95 -n 10 -M 10000). The total nonredundant contigs were used for virus identification by using VIBRANT (v1.2.1) (22) and verified using VirSorter (v1.0.6) (48). The putative viral contigs were classified using vConTACT2 (v0.9.19) (49) and predicted for their bacterial hosts based on CRISPR spacers. In the case of prophages, their bacterial hosts were gauged by their flanking bacterial sequences by using BLAST. Microbial contig (nonviral contig) classification was performed with Kraken2 (v2.0.8-beta). The personal reference metagenome (PRM) was constructed by merging annotated total viral and microbial contigs. The relative abundance of all species in a sample was calculated by directly mapping clean reads to the PRM using Bowtie2 (v2.3.5) (50) and normalized to 1 as follows:

$$\mathrm{abun}_i = \frac{C_i/L_i}{\sum_i^N (C_i/L_i)}$$

$$\mathrm{relative\ abundance}_A = \sum_i^n \mathrm{abun}_i$$

where $C_i$ is the number of reads mapping to contig $i$, $L_i$ is the length of contig $i$, $N$ is total contigs in the PRM, and $n$ is contigs classified into species $A$.

**MAG binning and taxonomical classification.** Filtered Illumina HiSeq reads of each sample were assembled using metaSPAdes. For each sample, contigs longer than 5,000 are clustered based on sequence using CD-HIT-EST (v4.8.1) with '-c 0.95'. MAG binning was performed using MetaWRAP with each sample's contigs and refinement with '-c 50 -x 10'. The total MAGs of 24 samples were clustered at the species level using dRep (v3.4.1) with '-sa 0.95 -nc 0.3 -comp 50 -con 10' to select representative MAGs. Taxonomic classification of representative MAGs was performed using GTDB-Tk (v2.1.1).

**Identification of AAA-metabolizing gene.** For FldH (phenyllactate dehydrogenase), FldBC (phenyl-lactate dehydratase subunits B and C), AcdA (acyl coenzyme A [acyl-CoA] dehydrogenase), and PorA (pyruvate:ferredoxin oxidoreductase A), which are essential for the metabolism of all three AAAs, we screened the UniProt Swiss-Prot database for these five protein sequences and constructed the HMM for each enzyme. Open reading frames (ORFs) on the PRM contig were predicted by Prodigal (version v2.6.3) (51), and we searched homologous proteins from the PRM in five enzyme HMM (cutoff, 1e−10). Homologous *fldB* and *fldC* genes located in the same contig and neighboring were retained. Mapping read statistics were performed using SAMtools (v0.1.19) (52) and BEDTools (v2.27.1) (53). Genes' or fragments' TPM was calculated as follows:

$$\mathrm{TPM}_j = \frac{(C_j/L_j) \times 10^6}{\sum_j^N (C_j/L_j)}$$

where $C_j$ is the number of reads mapping to gene/fragment $j$, $L_j$ is the length (kilobases) of gene/fragment $j$, and $N$ is total contigs in the PRM.

**SNP calling in crAss-99kb phage.** SNP calling in crAss-99kb phage sequences was performed on the SAM file of each sample generated from mapping clean reads to the PRM using Picard (version 2.26.2) and GATK (version 4.2.2.0) (54). The SNPs concentrate in a narrow 75-bp region in the coding sequence of the putative phage tail fiber protein. Those reads that completely covered the 75-bp region were clustered using CD-HIT-EST (version 4.8.1, option: -c 1 -n 10 -M 10000) (55) and recognized as several substrains of crAssphage-99kb. TPM was calculated as mentioned above.

**Metabolomics analysis of fecal samples.** The analysis of short-chain fatty acids and nontargeted metabolome of each fecal sample was performed using ultrahigh=performance liquid chromatography (UHPLC) on an Exion UPLC- quadrupole time of flight (QTOF) 5600 PLUS platform following routine operations by LipidALL Technologies (Beijing) Co., Ltd.

**Analysis of published metagenomic data set of an infant eczema cohort.** The published metagenomic data set from an infant eczema cohort was from the work of Murphy et al. (44). Infants in the group receiving placebos in the original cohort sampled from 1-year-olds and 2-year-olds were selected. The raw sequencing data were filtered according to the data quality control process mentioned above. The relative abundance of all viruses in a sample was calculated by directly mapping clean reads to the

GPD (http://ftp.ebi.ac.uk/pub/databases/metagenomics/genome_sets/gut_phage_database) (12) database using Bowtie2 (version 2.3.5) (50) and normalized to 1. Total viral genomes from GPD were annotated by using VIBRANT to identify the feature, such as lytic and lysogenic variants.

**Statistics and data visualizations.** All statistical analysis was performed by using R software v4.0.3. The significance of differences in abundance between groups was tested with the Mann-Whitney U test for each species with $P$ values adjusted by the Benjamini-Hochberg approach. FDR of $<0.05$ was considered to be significant. Correlations between species abundance and EASI score were evaluated with a Spearman rank correlation test. Several R packages were used for data visualization, including ggplot2, ggpubr, gggenes, and RColorBrewer. Visualization of homology between viral genomes was performed by using RectChr (https://github.com/BGI-shenzhen/RectChr).

**Ethical approval and consent to participate.** This study has been approved by the ethics committee of the Beijing Institute of Genomics, Chinese Academy of Sciences (Ethical Review Document no. 2019H009). This study gained informed consent from the parents/guardians for the collection of stool samples and information. Patients or the public were not involved in the design, conduct, reporting, or dissemination plans of our research.

**Data availability.** All data needed to evaluate the conclusions in the paper are present in the paper and/ or the supplemental material. The raw metagenome sequencing data reported in this paper have been deposited in the Genome Sequence Archive in BIG Data Center, Beijing Institute of Genomics (BIG), Chinese Academy of Sciences, under accession number CRA003594 at https://ngdc.cncb.ac.cn/gsa/browse/CRA003594. Additional data related to this paper may be requested from the authors.

## SUPPLEMENTAL MATERIAL

Supplemental material is available online only.
**SUPPLEMENTAL FILE 1**, PDF file, 4.5 MB.

## ACKNOWLEDGMENTS

Y.K. conceived and designed the project. Y.C. and J.Z. collected information and samples from the participant. Y.C. and J.C. performed DNA extraction. Y.C. and Q.M. processed and analyzed all the data. Y.C. performed the statistical analysis and visualization. Y.K. and J.Y. conceived and coordinated the project. Y.K., Y.C., and J.Y. contributed to the writing of the manuscript.

We declare that we have no competing interests.

This work was supported by the National Key Research and Development Program of China (2021YFC2301000) and the National Natural Science Foundation of China (31970568).

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
