## [Reviewer comments · Microbiology Spectrum]

Microbiology Spectrum

Strain-level Dynamics Reveals Regulatory Roles in Atopic Eczema by Gut Bacterial Phages

Yanan Chu, Qingren Meng, Jun Yu, Juan Zhang, Jing Chen, and Yu Kang

Corresponding Author(s): Yu Kang, Beijing Institute of Genomics Chinese Academy of Sciences

Review Timeline:

Submission Date:	November 9, 2022
Editorial Decision:	January 6, 2023
Revision Received:	February 21, 2023
Accepted:	March 6, 2023

Editor: Adelumola Oladeinde

Reviewer(s): Disclosure of reviewer identity is with reference to reviewer comments included in decision letter(s). The following individuals involved in review of your submission have agreed to reveal their identity: Jonas Grove (Reviewer #2)

Transaction Report:

DOI: <https://doi.org/10.1128/spectrum.04551-22>

January 6, 2023

Prof. Yu Kang
Beijing Institute of Genomics, Chinese Academy of Sciences, and China National Center for Bioinformation
Beijing
China

Re: Spectrum04551-22 (Strain-level Dynamics Reveals Regulatory Roles in Atopic Eczema by Gut Bacterial Phages)

Dear Prof. Yu Kang:

I have now received feedback from two experts in the field. Based on their recommendations and my own evaluation, the manuscript requires a substantial revision, and it is currently unsuitable for publication in Microbiology Spectrum. As you can see, both reviewers agreed that the results presented is interesting and the methods used are acceptable. Nonetheless, the reviewers were concerned about the English writing including grammatical and spelling errors. Furthermore, reviewer #1 suggested that the authors temper their inferential claims and focus on the data as it appears. Additionally, reviewer #2 mentioned that the methods used by the authors to generate PRM and estimate the abundance of MAGs need to be improved to avoid scrutiny and misinterpretation. Lastly, the authors should discuss the limitation of the study including the limitations of results derived from a single subject in this work. If you decide to resubmit the manuscript to Microbiology Spectrum, please ensure that the raw sequencing data is published and available to the reviewers,

Link Not Available

Sincerely,

Adelumola Oladeinde

Journals Department
Reviewer comments:

Reviewer #1 (Public repository details (Required)):

The raw sequencing data should be published and is noted as having been submitted to the GSA database to be released on publication.

Reviewer #1 (Comments for the Author):

The authors present a longitudinal case study of the fecal metagenome from a 2-year-old patient before, during, and after onset of atopic eczema. The authors assemble and annotate a personalized, de novo metagenome for the subject and use this personalized metagenome to assess longitudinal patterns of differential abundance of phages/prophages, their associate bacteria, and metabolites. Overall, the study presents data that support a hypothesis that a strain-specific interaction between phages and their host bacteria could be related to atopic eczema etiology and severity of symptoms. This hypothesis is then further evaluated using data from another, previously published longitudinal study. While the data from the case study is interesting and valuable, I list below several concerns with the manuscript as written.

Major comments:

1. The authors claim that the "Personal Reference Metagenomes (PRM)" methodology is a new approach. However, previously published, similar work has been done in the context of Metagenome-Assembled Genomes (MAGs) and MAG binning (Stewart et al 2018, "Assembly of 913 microbial genomes...", Nat Commun; Stewart et al. 2019, "MAGpy..." Bioinformatics). Given that the importance section in this manuscript focuses on the PRM approach, the existing area of bioinformatics methods should be addressed, and discussion should be included on how the PRM approach improves upon published methodology.
2. Additional discussion should be included on the limitations of results derived from a single subject in this work. While case studies from individual patients can be valuable, more discussion of the limitations of the findings is warranted than what is included currently in the manuscript.

Minor comments:

1. (Lines 118-125) "To construct PRM for individual..." and (Lines 137-139) "The final PRM..." - The authors claim that long read assemblies are satisfactory for accurate viral genome identification and taxonomic assignment but no numerical or summary data are provided to support the claim.
2. (Lines 156-159) "When mapping the short reads..." - The authors claim that an average short-read mapping rate indicates "high quality for our PRM assembly" that enables "exact estimation of the relative abundance of all viruses in our samples." It's not clear that a high mapping rate indicates high quality assemblies, particularly if the short read data were used in the construction of the assemblies; a different metric may be appropriate here. It's unclear what is meant by "exact estimation" when the data have been normalized by total sum scaling and/or RPKM.
3. (Lines 171-196) "Interestingly, such strain-level..." - The conclusion that the data display "a complex ecological relationship between viruses and their host" is overstated based on a sample size of one.
4. (Line 200, section title) "Upsurge of crAssphage substrains correlates..." - "Correlates" should be supported by a statistical analysis or measurement of correlation in the context of a longitudinal study; perhaps a different term would be appropriate here, since the authors analyzed differential abundance and not, for example, correlation with another measure like EASI, as was done for AAA metabolite concentration later in this section.
5. P-values, summary metrics, and statistical testing results listed in figures, tables, or figure/table captions (including supplementary material) should be reported in the main body text at the appropriate location if used to support statements made in the text.
6. Minor but noticeable grammatical and spelling errors occur throughout the work.

Reviewer #2 (Comments for the Author):

1. There are numerous grammatical and spelling errors that should be addressed before publication.
2. I find that the construction of the Personal Reference Metagenomes (PRM) could be a source of bias. The PRM were obtained using only two of the samples in the cohort and used to estimate genome abundance for every sample. Therefore, there could be viral and/or prokaryotic populations in the other samples that are not represented in the PRM, thus confounding the abundance results. An alternative way to do this, which would not require generating more sequence data, would be to assemble genomes from all of the samples to create an assembly for each sample. Then use the assembled contigs from each sample to produce the PRM in the same way that is described in the manuscript. I recognize that the lack of nanopore reads from the other samples in the cohort would make obtaining complete genomes from the other samples more difficult, but it may reveal populations that were missed using the current approach.
3. Performing metagenomic binning on the contigs obtained from each sample using a tool such as metabat2, concoct, or maxbin could yield more complete PRM.
4. Using a marker gene based taxonomic annotation tool, such as GTDBTK, could be a better option than using Kraken2, which assigns taxonomic labels based on kmer similarities.
5. Including completion and contamination metrics using checkM and checkV on the PRM could provide higher confidence in the analysis.
6. Adding more specifics on how the abundance was estimated to the methods section would increase confidence in the results

of the analysis. The figure axis are labelled as RPKM, but the methods section does not thoroughly explain how the abundance values were obtained. Additionally, explaining why RPKM was used could benefit the analysis. Generally, TPM is the favored normalization technique when comparing abundances between samples.

Staff Comments:

Preparing Revision Guidelines

Please return the manuscript within 60 days; if you cannot complete the modification within this time period, please contact me. If you do not wish to modify the manuscript and prefer to submit it to another journal, please notify me of your decision immediately so that the manuscript may be formally withdrawn from consideration by Microbiology Spectrum.

The authors present a longitudinal case study of the fecal metagenome from a 2-year-old patient before, during, and after onset of atopic eczema. The authors assemble and annotate a personalized, *de novo* metagenome for the subject and use this personalized metagenome to assess longitudinal patterns of differential abundance of phages/prophages, their associated bacteria, and metabolites. Overall, the study presents data that support a hypothesis that a strain-specific interaction between phages and their host bacteria could be related to atopic eczema etiology and severity of symptoms. This hypothesis is then further evaluated using data from another, previously published longitudinal study. While the data from the case study is interesting and valuable, I list below several concerns with the manuscript as written.

Major comments:

1. The authors claim that the “Personal Reference Metagenomes (PRM)” methodology is a new approach. However, previously published, similar work has been done in the context of Metagenome-Assembled Genomes (MAGs) and MAG binning (Stewart et al 2018, “Assembly of 913 microbial genomes...”, *Nat Commun*; Stewart et al. 2019, “MAGpy...” *Bioinformatics*). Given that the importance section in this manuscript focuses on the PRM approach, the existing area of bioinformatics methods should be addressed, and discussion should be included on how the PRM approach improves upon published methodology.
2. Additional discussion should be included on the limitations of results derived from a single subject in this work. While case studies from individual patients can be valuable, more discussion of the limitations of the findings is warranted than what is included currently in the manuscript.

Minor comments:

1. (Lines 118-125) “To construct PRM for individual...” and (Lines 137-139) “The final PRM...” – The authors claim that long read assemblies are satisfactory for accurate viral genome identification and taxonomic assignment but no numerical or summary data are provided to support the claim.
2. (Lines 156-159) “When mapping the short reads...” – The authors claim that an average short-read mapping rate indicates “high quality for our PRM assembly” that enables “exact estimation of the relative abundance of all viruses in our samples.” It’s not clear that a high mapping rate indicates high quality assemblies, particularly if the short read data were used in the construction of the assemblies; a different metric may be appropriate here. It’s unclear what is meant by “exact estimation” when the data have been normalized by total sum scaling and/or RPKM.
3. (Lines 171-196) “Interestingly, such strain-level...” – The conclusion that the data display “a complex ecological relationship between viruses and their host” is overstated based on a sample size of one.
4. (Line 200, section title) “Upsurge of crAssphage substrains correlates...” – “Correlates” should be supported by a statistical analysis or measurement of correlation in the context of a longitudinal study; perhaps a different term would be appropriate here, since the authors analyzed

differential abundance and not, for example, correlation with another measure like EASI, as was done for AAA metabolite concentration later in this section.

5. P-values, summary metrics, and statistical testing results listed in figures, tables, or figure/table captions (including supplementary material) should be reported in the main body text at the appropriate location if used to support statements made in the text.
6. Minor but noticeable grammatical and spelling errors occur throughout the work.

Point-by-point response to the reviewers

Our response is in blue.

Reviewer comments:

Reviewer #1 (Public repository details (Required)):

The raw sequencing data should be published and is noted as having been submitted to the GSA database to be released on publication.

Response: Thanks for the suggestion. The raw sequence data reported in our manuscript have been deposited in a publicly accessible repository, the Genome Sequence Archive in National Genomics Data Center, China National Center for Bioinformatics <https://ngdc.cnbc.ac.cn/gsa> (GSA: CRA003594). To facilitate the manuscript review, you can access the data from the temporary link: <https://ngdc.cnbc.ac.cn/gsa/s/Jd2K2S7i> before Mar 17. We hope to keep the data private until acceptance. We have noted it in the "Data Availability" Section of the revised manuscript.

Reviewer #1 (Comments for the Author):

The authors present a longitudinal case study of the fecal metagenome from a 2-year-old patient before, during, and after onset of atopic eczema. The authors assemble and annotate a personalized, de novo metagenome for the subject and use this personalized metagenome to assess longitudinal patterns of differential abundance of phages/prophages, their associate bacteria, and metabolites. Overall, the study presents data that support a hypothesis that a strain-specific interaction between phages and their host bacteria could be related to atopic eczema etiology and severity of symptoms. This hypothesis is then further evaluated using data from another, previously published longitudinal study. While the data from the case study is interesting and valuable, I list below several concerns with the manuscript as written.

Major comments:

1. The authors claim that the "Personal Reference Metagenomes (PRM)" methodology is a new approach. However, previously published, similar work has been done in the context of Metagenome-Assembled Genomes (MAGs) and MAG binning (Stewart et al 2018, "Assembly of 913 microbial genomes...", Nat Commun; Stewart et al. 2019, "MAGpy..." Bioinformatics). Given that the importance section in this manuscript focuses on the PRM approach, the existing area of bioinformatics methods should be addressed, and discussion should be included on how the PRM approach improves upon published methodology.

Response: Thank you for your insightful comment. In this work, to disentangle the role of gut virome in eczema and its relationship to gut microbes, we have to calculate the relative abundance of each virus to the entire microbiome besides the virome, for which a non-redundant reference is required through read-mapping. Actually, before the PRM based on the long-read sequencing approach, we tried the binning strategy to obtain assembled the reference genomes of both bacteria and viruses, i.e., Metagenome-Assembled Genomes (MAGs) reconstructions from shotgun metagenomic sequence data. However, the results are not satisfying for our particular sample set, *i.e.*, dynamic samples from a single patient. The binning process lost many sequences even though we pooled all samples together, including the crAssphage. Moreover, the pooling of many samples from the same subject, where the microbial composition is quite similar among samples, led to considerable redundancy in the reference, which would substantially underestimate the abundance. Although the current de-redundancy tools, such as dRep, can correctly trim the redundancy of bacteria MAGs binned by MetaBAT, CONCOCT, or MaxBin, the binning algorithm for viruses is so different and more challenging that there are currently few tools available for virus binning (Kristopher Kieft et al, Nucleic Acids Research, 2022, “vRhyme”; Joachim Johansen et al, Nature Communications, 2021, “PHAMB”), and no tool can bin virus and bacteria MAGs from metagenomes sequencing data at the same time without training by VLP sequencing data.

For achieving the low-redundant Personal Reference Metagenome (PRM), long-read sequencing of representative samples is an alternative approach, which avoids high redundancy in the reference and recovers more viral genome contigs due to the long contigs it assembled. To illustrate the appropriateness of our PRM methods for the longitudinal samples from a single subject, we compared it with the binning approach and listed the results in the following table:

Table R1. The comparison between PRM and MAG reference

	PRF(5k)	MAG(5K)
The contigs' number	5,431	9,753
N50 length	71,829	74,150
N10 length	677,509	302,150
No. of contigs >500k	27	14
No. of contigs >1M	5	1
The max length	2,796,291	1,415,749
The full length	140,768,589	294,898,180
VIBRANT predicts virus		
No. of circular/complete virus	9	3

Virus total Length	479,560	109,496
High-quality virus	17	26
High-quality virus total length	1,034,433	1,346,934
Medium-low quality virus	217	198
No. of total virus fragments	234	224
Mapping rate	99.10±0.52%	85.55±0.04%

We performed MAG binning and taxonomical classification on the assembly of 24 samples' shotgun sequencing data as follows:

Filtered Illumina HiSeq reads of each sample were assembled using metaSPAdes. For each sample, contigs longer than 5k are clustered based on sequence using CD-HIT-EST (v4.8.1) with '-c 0.95'. Perform binning using MetaWRAP with each sample's contigs and refinement with '-c 50 -x 10' to yield MAGs. The total MAGs of 24 samples were clustered at the species level using dRep (v3.4.1) with '-sa 0.95 -nc 0.3 -comp 50 -con 10' to select representative MAGs. Taxonomic annotation of representative MAGs was performed using GTDB-Tk (v2.1.1).

The comparison shows that the PRM annotated more viruses than the binned MAGs after pooling all 24 samples, especially for the complete and circular virus, which depends on the length of contigs. Furthermore, the crAssphage we focused on was missed from the MAG-based PRM, and the lower mapping rate to it indicated a considerable proportion of sample DNA material was not represented in reference. Therefore, we think the PRM we proposed is more appropriate for dynamic samples from a single subject. And we added this comparison in the revised manuscript as follows and quoted the relevant articles.

Lines 168-178, "Metagenome-Assembled Genomes (MAGs) reconstructions from shotgun metagenomic sequence data is another approach to obtaining the PRM that needs no extra long-read sequencing (24-26). Here, we also tried this strategy by binning the short-read assembled contigs of each sample using MetaWRAP. Then we pooled all the bins from the 24 samples and removed redundancy using dRep to obtain the MAG-based PRM. Compared with our long-read-based PRM, the MAG-based PRM performs worse in virus identification, especially for the circular or high-quality genomes, due to the relatively short length of its contigs (Supplementary Table S5). The final MAG-based PRM also lost many sequences even after pooling bins from all the samples due to the trimmed contigs in the binning process, which leads to a significantly lower mapping rate of 85.55±0.04%, indicating its poorer representativeness."

2. Additional discussion should be included on the limitations of results derived

from a single subject in this work. While case studies from individual patients can be valuable, more discussion of the limitations of the findings is warranted than what is included currently in the manuscript.

Response: Thank you for your valuable suggestions. We added this section in the revised manuscript as follows.

Lines 349-360, " Although exciting findings from the single case, definite conclusions on the role of virome, especially crAssphages, in maintaining the normal immune status of a human host are far from being established. In most of the cross-sectional studies on virome, the huge strain diversity in virome profiles among human individuals makes it more difficult to compare a given virus across subjects, and the assemblies based on short-read sequences impair the annotation of virome. Thus, we cannot fully validate our findings on the roles of the crAssphages and other phages in infant eczema due to the lack of public sequencing data of longitudinal observation on gut virome, especially those of long-read sequencing. However, the complex strain-sensitive phage-host ecological relationship and the potential effects of virome alteration on human healthy we observed in the single case highlights the importance of longitudinal studies and the merit of long-read-based sequencing in deciphering the pathogenesis of gut phages in various human diseases."

Minor comments:

1. (Lines 118-125) "To construct PRM for individual..." and (Lines 137-139) "The final PRM..." - The authors claim that long read assemblies are satisfactory for accurate viral genome identification and taxonomic assignment but no numerical or summary data are provided to support the claim.

Response: Thank you for the valuable comment. As in our response to the 1st major comment, we have performed the comparison in the viruses annotated from the PRM and the MAG-based reference. Table R1 presented the numerical comparison, which supports the appropriation of PRM for the dynamic studies of virome. We have added this table to the revised manuscript as you suggested.

Lines 168-178, "Metagenome-Assembled Genomes (MAGs) reconstructions from shotgun metagenomic sequence data is another approach to obtaining the PRM that needs no extra long-read sequencing. Here, we also tried this strategy by binning the short-read assembled contigs of each sample using MetaWRAP. Then we pooled all the bins from the 24 samples and removed redundancy using dRep to obtain the MAG-based PRM. Compared with our long-read-based PRM, the MAG-based PRM performs worse in virus identification, especially for the circular

or high-quality genomes, due to the relatively short length of its contigs (Supplementary Table S5). The final MAG-based PRM also lost many sequences even after pooling bins from all the samples due to the trimmed contigs in the binning process, which leads to a significantly lower mapping rate of $85.55 \pm 0.04\%$, indicating its poorer representativeness."

2. (Lines 156-159) "When mapping the short reads..." - The authors claim that an average short-read mapping rate indicates "high quality for our PRM assembly" that enables "exact estimation of the relative abundance of all viruses in our samples." It's not clear that a high mapping rate indicates high quality assemblies, particularly if the short read data were used in the construction of the assemblies; a different metric may be appropriate here. It's unclear what is meant by "exact estimation" when the data have been normalized by total sum scaling and/or RPKM.

Response: Thank you for the insightful comments. We agree that the wording in these sentences is not accurate. We mean that our PRM can represent the microbial DNA in the sample very well when the mapping rate reaches as high as 99%. As for the abundance, we want to highlight that the relative abundance of each virus in the whole microbial flora than in the virome can be more accurately evaluated by mapping the reads to the reference than marker-gene-based abundance evaluation, especially for the relative abundance between viruses and bacteria. We have rephrased these sentences in the revised manuscript to be more precise and accurate as follows.

Lines 161-162, "Such an unprecedentedly mapping rate indicates PRM can represent the microbial DNA in the samples with negligible loss."

3. (Lines 171-196) "Interestingly, such strain-level..." - The conclusion that the data display "a complex ecological relationship between viruses and their host" is overstated based on a sample size of one.

Response: Thank you for the comment. We have rephrased it in the revised manuscript as follows.

Lines 184-186, "Interesting strain-level phage-host dynamics also implies complex ecological relationship among viruses and their bacterial hosts..."

Lines 349-360, "Although exciting findings from the single case, definite conclusions on the role of virome, especially crAssphages, in maintaining the normal immune status of a human host are far from being established. In most of the cross-sectional studies on virome, the huge strain diversity in virome profiles among human

individuals makes it more difficult to compare a given virus across subjects, and the assemblies based on short-read sequences impair the annotation of virome. Thus, we cannot fully validate our findings on the roles of the crAssphages and other phages in infant eczema due to the lack of public sequencing data of longitudinal observation on gut virome, especially those of long-read sequencing. However, the complex strain-sensitive phage-host ecological relationship and the potential effects of virome alteration on human healthy we observed in the single case highlights the importance of longitudinal studies and the merit of long-read-based sequencing in deciphering the pathogenesis of gut phages in various human diseases."

4. (Line 200, section title) "Upsurge of crAssphage substrains correlates..." - "Correlates" should be supported by a statistical analysis or measurement of correlation in the context of a longitudinal study; perhaps a different term would be appropriate here, since the authors analyzed differential abundance and not, for example, correlation with another measure like EASI, as was done for AAA metabolite concentration later in this section.

Response: Thank you for your valuable comment and suggestion. We complement the correlation analysis in the revised version. Table R2 and Figure R1 show the correlations between phages abundance and EASI score, evaluated with a Spearman's rank correlation test ($|\rho| > 0.5$ and $P < 0.01$). The lytic crAssphage_99kb and circular_phage_158kb show the strongest negative correlations to the EASI score, whereas a variety of prophages exhibit strong positive correlations. We have added this table and figure to the revised manuscript.

Table R2. Correlations between phages abundance and EASI score. ($|\rho| > 0.5$ and $P < 0.01$)

EASI	phage	type	ρ	P-value	correlations
EASI	unassigned_circular_phage_158kb	lytic	-0.59521	0.002153	negative
EASI	crAssphage_99kb	lytic	-0.52887	0.007881	negative
EASI	unassigned_proPhage_55kb	lysogenic	0.530764	0.007621	positive
EASI	unassigned_proPhage_62kb	lysogenic	0.623648	0.001129	positive
EASI	unassigned_proPhage_38kb	lysogenic	0.64829	0.000613	positive
EASI	Streptomyces_proPhage_57kb	lysogenic	0.651134	0.000569	positive

Figure R1. Correlations between phages abundance and EASI score

5. P-values, summary metrics, and statistical testing results listed in figures, tables, or figure/table captions (including supplementary material) should be reported in the main body text at the appropriate location if used to support statements made in the text.

Response: Thank you for the suggestion. We complemented them in revised manuscript as you suggested.

6. Minor but noticeable grammatical and spelling errors occur throughout the work.

Response: Thank you for the critique. We have combed through any grammar and spelling errors in the revised manuscript and corrected them. We are sorry for these mistakes.

Reviewer #2 (Comments for the Author):

1. There are numerous grammatical and spelling errors that should be addressed before publication.

Response: Thank you for the critique. We have combed through any grammar and

spelling errors in the revised manuscript and corrected them. We are sorry for these mistakes.

2. I find that the construction of the Personal Reference Metagenomes (PRM) could be a source of bias. The PRM were obtained using only two of the samples in the cohort and used to estimate genome abundance for every sample. Therefore, there could be viral and/or prokaryotic populations in the other samples that are not represented in the PRM, thus confounding the abundance results. An alternative way to do this, which would not require generating more sequence data, would be to assemble genomes from all of the samples to create an assembly for each sample. Then use the assembled contigs from each sample to produce the PRM in the same way that is described in the manuscript. I recognize that the lack of nanopore reads from the other samples in the cohort would make obtaining complete genomes from the other samples more difficult, but it may reveal populations that were missed using the current approach.

Response: Thank you for the insightful comment. In most metagenomic studies, which are cross-sectional, reference genomes assembled from one sample cannot represent that of others due to the great diversity between individuals. However, in our study, all the samples were from a single patient within a short period, and the components of his gut flora are supposed to be relatively stable while the abundance might fluctuate. The results show that the PRM derived from just two samples can well represent the components of other samples since the read mapping rate achieved $99.1 \pm 0.5\%$, which means almost all the DNA sequences in the other samples can be found in the PRM.

Pooling all samples is a common approach in assembling metagenome but inappropriate in our case because pooling samples from the same subject can result in overwhelming redundancy, which is not easy to remove completely using the cd-hit tool. The redundant sequences containing the same repeating segments might undervalue the abundance.

3. Performing metagenomic binning on the contigs obtained from each sample using a tool such as metabat2, concoct, or maxbin could yield more complete PRM.

Response: Thank you for the valuable suggestion. In this work, we focused on the relative abundance of each virus to the entire microbiome instead of that to the virome, for which a non-redundant reference is required through read-mapping. Actually, before the PRM based on the long-read sequencing approach, we tried the binning strategy to obtain the assembled reference genomes of both bacteria and

viruses, i.e., Metagenome-Assembled Genomes (MAGs) reconstructions from shotgun metagenomic sequence data. However, we found that the binning process not only lost many sequences, including the crAssphage, even though we pooled all samples together, but also led to considerable redundancy in the reference. Although the current de-redundancy tools, such as dRep, can correctly trim the redundancy of bacteria MAGs binned by MetaBAT, CONCOCT, or MaxBin, the binning algorithm for viruses is so different and more challenging that there are currently few tools available for virus binning (Kristopher Kieft et al, Nucleic Acids Research, 2022, “vRhyme”; Joachim Johansen et al, Nature Communications, 2021, “PHAMB”), and no tool can bin virus and bacteria MAGs from metagenomes sequencing data at the same time without training by VLP sequencing data.

For achieving the low-redundant Personal Reference Metagenomes (PRM), long-read sequencing of representative samples is an alternative approach, which avoids high redundancy in the reference and recovers more viral genome contigs due to the long contigs it assembled. With the rapid cost decline of long-read sequencing, the PRM method is more affordable than ever. To illustrate the appropriateness of our PRM methods for the longitudinal samples from a single subject, we compared it with the binning approach and listed the results in the following table:

Table R3. The comparison between PRM and MAG reference

	PRF(5k)	MAG(5K)
The contigs' number	5,431	9,753
N50 length	71,829	74,150
N10 length	677,509	302,150
No. of contigs >500k	27	14
No. of contigs >1M	5	1
The max length	2,796,291	1,415,749
The full length	140,768,589	294,898,180
VIBRANT predicts virus		
No. of circular/complete virus	9	3
Virus total Length	479,560	109,496
High-quality virus	17	26
High-quality virus total length	1,034,433	1,346,934
Medium-low quality virus	217	198
No.of total virus fragments	234	224
Mapping rate	99.10±0.52%	85.55±0.04%

We performed MAG binning and taxonomical classification on the assembly of 24 samples' shotgun sequencing data as follows:

Filtered Illumina HiSeq reads of each sample were assembled using metaSPAdes. For each sample, contigs longer than 5k are clustered based on sequence using CD-HIT-EST (v4.8.1) with '-c 0.95'. Perform binning using MetaWRAP with each sample's contigs and refinement with '-c 50 -x 10' to yield MAGs. The total MAGs of 24 samples were clustered at the species level using dRep (v3.4.1) with '-sa 0.95 -nc 0.3 -comp 50 -con 10' to select representative MAGs. Taxonomic annotation of representative MAGs was performed using GTDB-Tk (v2.1.1).

The comparison shows that the PRM annotated more viruses than the binned MAGs after pooling all 24 samples, especially for the complete and circular virus, which depends on the length of contigs. Furthermore, the crAssphage we focused on was missed from the MAG-based PRM, and the lower mapping rate to it indicated a considerable proportion of sample DNA material was not represented in reference. Therefore, we think the PRM we proposed is more appropriate for dynamic samples from a single subject. And we added this comparison in the revised manuscript as follows.

Lines 168-178, "Metagenome-Assembled Genomes (MAGs) reconstructions from shotgun metagenomic sequence data is another approach to obtaining the PRM that needs no extra long-read sequencing (24-26). Here, we also tried this strategy by binning the short-read assembled contigs of each sample using MetaWRAP. Then we pooled all the bins from the 24 samples and removed redundancy using dRep to obtain the MAG-based PRM. Compared with our long-read-based PRM, the MAG-based PRM performs worse in virus identification, especially for the circular or high-quality genomes, due to the relatively short length of its contigs (Supplementary Table S5). The final MAG-based PRM also lost many sequences even after pooling bins from all the samples due to the trimmed contigs in the binning process, which leads to a significantly lower mapping rate of $85.55 \pm 0.04\%$, indicating its poorer representativeness."

4. Using a marker gene based taxonomic annotation tool, such as GTDBTK, could be a better option than using Kraken2, which assigns taxonomic labels based on kmer similarities.

Response: Thank you for the suggestion. GTDB-Tk is a software designed for assigning objective taxonomic classifications to bacterial genomes or MAGs. As the contigs in PRM were not binned into MAGs, we selected Kraken for taxonomic annotation for our contigs in PRM according to the Critical Assessment of Metagenome Interpretation (CAMI), which verified the best performance of Kraken2 in contigs annotation (Fernando Meyer et al., Nature Methods 2022). Furthermore,

we also tried Contig Annotation Tool (CAT) (von Meijenfeldt FAB et al. Genome Biology, 2019) for the taxonomic classification of contigs based on gene homology search. Few contigs can be classified into species (Table R4), far less than that with Kraken2.

Table R4.

	Contigs classified into species		
	contigs%	length %	Number of bacterial species
Kraken2	0.858773706	0.946239	473
CAT	0.136943264	0.076841	93

5. Including completion and contamination metrics using checkM and checkV on the PRM could provide higher confidence in the analysis.

Response: Thank you for the valuable suggestion. We assessed the completeness and contamination of virus contigs/genomes using checkV. The quality evaluation of most viruses mentioned in this study, as shown in Table R5, is consistent with that evaluated using VIBRANT, and these results were supplemented in the revised manuscript.

As the bacterial contigs are not binned in the PRM, we do not perform CheckM to assess their completeness and contamination.

Table R5. The list of phage strains identified and evaluated by checkV.

VIBRANT					checkV		
phage/ prophage	name	length	type	quality	checkv_quality	completeness	contamination
	[Clostridium]_innocuum_phage_63kb	62677	lytic	high	high	90.21	0
	Clostridium_phage_14kb	13989	lytic	low	low	25.41	0
	Clostridium_phage_40kb	39812	lytic	complete circular	complete	100	0
	Clostridium_phage_42kb	42434	lytic	medium	high	100	9.12
	Coliphage_phiX174	5487	lytic	complete circular	complete	100	0
	unassigned_phage_49kb	49067	lytic	high	medium	62.6	0
	unassigned_phage_66kb	65625	lytic	high	medium	83.41	0
	crAssphage_99kb	99416	lytic	complete circular	high	93.52	0
phage	unassigned_circular_phage_158kb	157775	lytic	complete circular	high	100	0
	unassigned_circular_phage_6kb	6108	lytic	complete circular	complete	100	0
	Vibrio_phage_26kb	26428	lytic	low	low	48.36	0
	Vibrio_phage_47kb	47009	lytic	complete circular	complete	100	0
	Faecalibacterium_phage_54kb	54586	lysogenic	medium	high	92.08	0
	Faecalibacterium_phage_55kb	54966	lysogenic	medium	high	100	0
	unassigned_phage_34kb	33770	lysogenic	high	medium	76.57	21.95
	unassigned_circular_phage_39kb	38878	lysogenic	complete circular	complete	100	0
	unassigned_circular_phage_42kb	42481	lysogenic	complete circular	complete	100	0
	unassigned_circular_phage_43kb	42594	lysogenic	complete circular	complete	100	0
prophage	[Clostridium]_innocuum_proPhage_66kb	65997	lysogenic	high	high	100	26.89
	Faecalibacterium_prausnitzii_proPhage_32kb	32153	lysogenic	high	high	92.16	0
	Faecalibacterium_prausnitzii_proPhage_63kb	62557	lysogenic	high	medium	84.42	16.71

Faecalibacterium_prausnitzii_proPhage_90kb	89943	lysogenic	high	high	92.64	49.52
Faecalibacterium_proPhage_38kb	38235	lysogenic	medium	high	98.99	0
Faecalibacterium_proPhage_55kb	54816	lysogenic	medium	medium	69.28	23.81
Lactobacillus_proPhage_Lc-Nu_24kb	24142	lysogenic	medium	medium	61.12	0
Lactobacillus_proPhage_PLE3_46kb	45729	lysogenic	high	high	95.99	13.18
Ruminococcus_proPhage_71kb	70841	lysogenic	high	complete	100	23.98
Streptomyces_proPhage_57kb	57169	lysogenic	high	high	98.88	0
unassigned_proPhage_55kb	54952	lysogenic	high	high	98.99	10.07
unassigned_proPhage_38kb	37899	lysogenic	high	high	100	0
unassigned_proPhage_62kb	62402	lysogenic	high	high	100	10.31

6. Adding more specifics on how the abundance was estimated to the methods section would increase confidence in the results of the analysis. The figure axis are labelled as RPKM, but the methods section does not thoroughly explain how the abundance values were obtained. Additionally, explaining why RPKM was used could benefit the analysis. Generally, TPM is the favored normalization technique when comparing abundances between samples.

Response: Thank you for the constructive suggestion. TPM is truly more appropriate than RPKM, and we calculated the TPM of each AAA-metabolizing gene and replaced the RPKM-based evaluation and comparison results with the new ones in the revised manuscript. The method of calculating TPM was added to the Methods section as follows.

$$TPM_j = \frac{(C_j/L_j) \times 10^6}{\sum_j^N (C_j/L_j)}$$

C_j , the number of reads mapping to gene/fragment j ; L_j , the length of gene/fragment j ; N , total contigs in PRM.

The “relative abundance” of each bacterial species and virus in the original manuscript is calculated as follows, and we add this section in Methods of the revised manuscript.

$$abun_i = \frac{C_i/L_i}{\sum_i^N (C_i/L_i)}$$

$$Relative\ Abundance_A = \sum_i^n abun_i$$

C_i , the number of reads mapping to contig i ; L_i , the length of contig i ; N , total contigs in RPM. n , contigs classified into species A.

March 6, 2023

Prof. Yu Kang
Beijing Institute of Genomics Chinese Academy of Sciences
Beijing
China

Re: Spectrum04551-22R1 (Strain-level Dynamics Reveals Regulatory Roles in Atopic Eczema by Gut Bacterial Phages)

Dear Prof. Yu Kang:

Your manuscript has been accepted, and I am forwarding it to the ASM Journals Department for publication. However, there are some comments from the editor that you must address before final publication, please make the necessary adjustments.

(1) There are still some minor grammatical errors that should be addressed throughout.

For instance, Lines 290 - 294. Here, the authors tried to validate their results by analyzing previously published metagenomic datasets. Therefore, the sentences should be written in past tense or part participle.
line 293 - what does sub-optimally mean here? I don't think it is necessary. The sentence can begin with "Two of the studies"
line 294 - collected not collect
line 294 - only one "had" not has
line 295 - the phrase "which metagenomic and metadata are fully released" is not clear. Are there some data that was "partially" released? I will re-write the sentence to say - " Two of the studies collected cross-sectional cohorts, and only one (45) had longitudinal metagenomic and metadata for gut samples from each subject.

Line 296 : fecal samples "were" not "are."

line 297 : " was" not specified.

(2) Figure S1; Change "Virual" to "Viral".

You will be notified when your proofs are ready to be viewed.

Sincerely,

Adelumola Oladeinde
Editor, Microbiology Spectrum
